# MILD-Net: Minimal Information Loss Dilated Network for Gland Instance Segmentation in Colon Histology Images

**Simon Graham**[1,2]    **Hao Chen**[3]    **Qi Dou**[3]    **Pheng Ann-Heng**[3]    **Nasir M. Rajpoot**[2,4,5]

[1]Mathematics for Real-World Systems Centre for Doctoral Training, University of Warwick, UK
[2]Department of Computer Science, University of Warwick, UK
[3]Department of Computer Science and Engineering, The Chinese University of Hong Kong, China
[4]Department of Pathology, University Hospitals Coventry and Warwickshire, Coventry, UK
[5]The Alan Turing Institute, London, UK
s.graham.1@warwick.ac.uk

## Abstract

The analysis of glandular morphology within colon histopathology images is a crucial step in determining the stage of colon cancer. Despite the importance of this task, manual segmentation is laborious, time-consuming and can suffer from subjectivity among pathologists. The rise of computational pathology has led to the development of automated methods for gland segmentation that aim to overcome the challenges of manual segmentation. However, this task is non-trivial due to the large variability in glandular appearance and the difficulty in differentiating between certain glandular and non-glandular histological structures. Furthermore, within pathological practice, a measure of uncertainty is essential for diagnostic decision making. For example, ambiguous areas may require further examination from numerous pathologists. To address these challenges, we propose a fully convolutional neural network that counters the loss of information caused by max-pooling by re-introducing the original image at multiple points within the network. We also use atrous spatial pyramid pooling with varying dilation rates for resolution maintenance and multi-level aggregation. To incorporate uncertainty, we introduce random transformations during test time for an enhanced segmentation result that simultaneously generates an uncertainty map, highlighting areas of ambiguity. We show that this map can be used to define a metric for disregarding predictions with high uncertainty. The proposed network achieves state-of-the-art performance on the GlaS challenge dataset, as part of MICCAI 2015, and on a second independent colorectal adenocarcinoma dataset.

## 1  Introduction

Colorectal cancer is the third most commonly occurring cancer in men and the second most commonly occurring cancer in women, where approximately 95% of all colorectal cancers are adenocarcinomas [7]. Colorectal adenocarcinoma develops in the lining of the colon or rectum, which makes up the large intestine and is characterised by glandular formation. Histological examination of the glands, most frequently with the Hematoxylin & Eosin (H&E) stain, is routine practice for assessing the differentiation of the cancer within colorectal adenocarcinoma. Within well differentiated cases, above 95% of the tumour is gland forming [7], whereas in poorly differentiated cases, typical glandular appearance is lost. Within the top row of Figure 1, (a) shows a healthy case, (b) shows a moderately differentiated tumour and (c) shows a poorly differentiated tumour. We observe the loss of glandular formation as the grade of cancer increases.

1st Conference on Medical Imaging with Deep Learning (MIDL 2018), Amsterdam, The Netherlands.

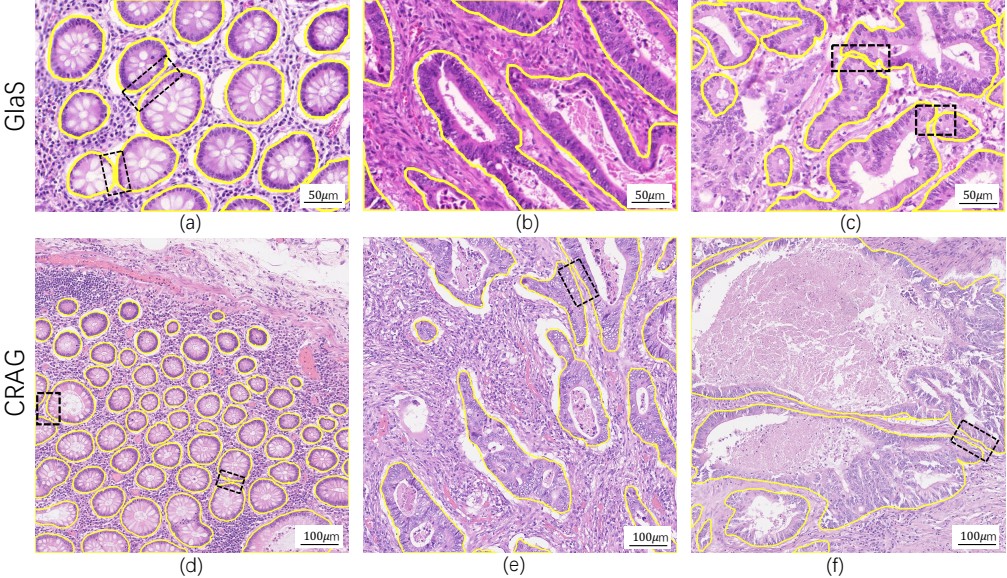

Figure 1: (a-c) Example images from the GlaS dataset [17]. (d-f) Example images from the CRAG dataset. All images displayed have overlaid boundary ground truth as annotated by an expert pathologist and are at 20× magnification. (a) and (d) show healthy glands, whereas the other images contain malignant glands. Black boxes highlight clustered glands.

There is a growing trend towards a digitised pathology workflow, where digital images are acquired from glass histology slides using a scanning device. The advent of digital pathology has led to a rise in computational pathology, where algorithms are implemented to assist pathologists in diagnostic decision making. In routine pathological practice, accurate segmentation of structures such as glands and nuclei are of crucial importance because their morphological properties can assist a pathologist in assessing the degree of malignancy [6, 10, 18]. With the advent of computational pathology, digitised histology slides are being leveraged such that pathological segmentation tasks can be completed in an objective manner. In particular, automated gland segmentation within H&E images can enable pathologists to extract vital morphological features from large scale histopathology images, that would otherwise be very time-consuming.

Most of the previous literature focused on the hand-crated features for histopathological image analysis [9]. Recently, deep learning achieved great success on image recognition tasks with powerful feature representation. For example, U-Net achieved excellent performance on the gland segmentation task [16]. To further improve the gland instance segmentation performance, Chen et al. presented a deep contour-aware network by formulating an explicit contour loss function in the training process and achieved the best performance during the 2015 MICCAI Gland Segmentation (GlaS) on-site challenge [4, 17]. In addition, a framework was proposed in [19] by fusing complex multichannel regional and boundary patterns with side supervision for gland instance segmentation. This work was extended in [20] to incorporate additional bounding box information for an enhanced performance. A Multi-Input-Multi-Output network (MIMO-Net) was presented for gland segmentation in [15] and achieved the state-of-the-art performance.

However, automated gland segmentation remains a challenging task due to several important factors. First, a high resolution level is needed for precise delineation of glandular boundaries, that is important when extracting morphological measurements. Next, glands vary in their size and shape, especially as the grade of cancer increases. Finally, there are areas of uncertainty within the images that current methods do not take into account. For example, areas of dense nuclei and lumenal areas have high uncertainty because of their similar appearance in both classes.

In this paper we propose a minimal information loss dilated network that aims to solve the key challenges posed by automated gland segmentation. During feature extraction, we introduce 'minimal information loss units', where we incorporate the original downsampled image into the residual unit

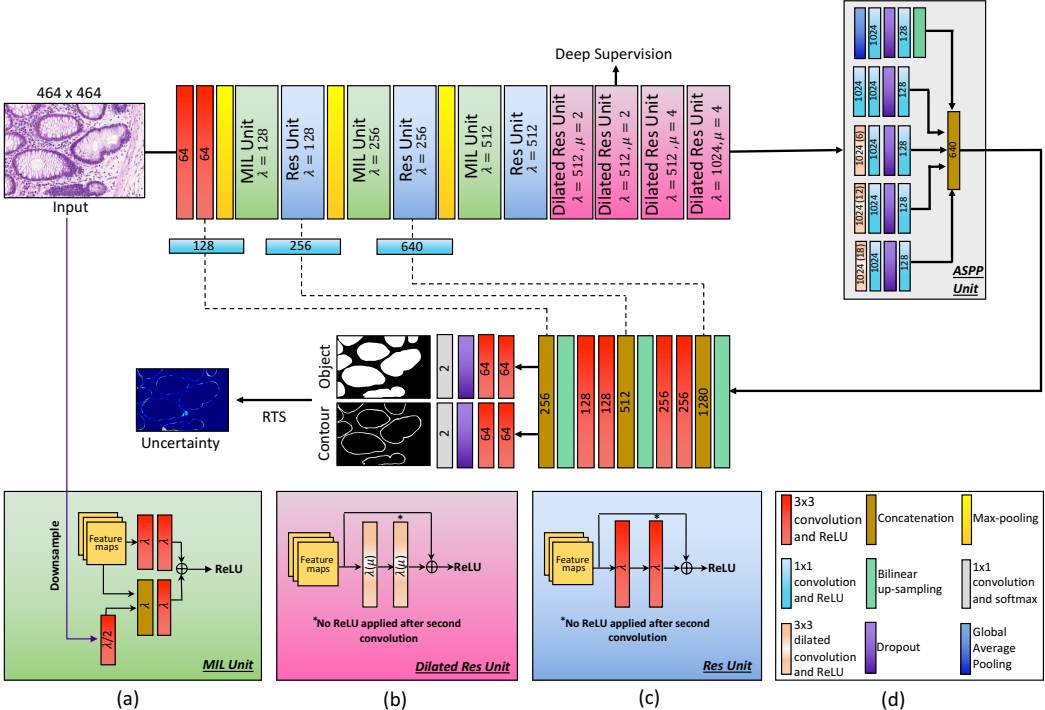

Figure 2: The overall framework of the proposed method. (a-c) Illustration of the varying residual units. (d) Key showing the components within the framework. The value within the centre of all convolutional, dilated convolutional and concatenation operators denotes the output depth. The value in brackets within the dilated convolutional operators denotes the dilation rate. $\lambda$ and $\mu$ refer to the output depth and dilation rate respectively within the given residual units where they are described. RTS refers to random transformation sampling

after max-pooling. This, alongside dilated convolution, helps retain maximal information that is essential for segmentation, particularly at the glandular boundaries. We use atrous spatial pyramid pooling for multi-level aggregation that is essential when segmenting glands of varying shapes and sizes. Despite deep neural networks achieving state-of-the-art performance in current segmentation tasks, they fail to take into account the uncertainty of a decision. During test time, we apply random transformations to the input images as a method of generating the predictive distribution. This leads to a superior segmentation result and allows us to observe areas of uncertainty that may be clinically informative. Furthermore, we use this measure of uncertainty to rank images that should be prioritised for pathologist annotation. Our proposed framework can be trained end to end, with one minimal information loss dilated feature extraction network. Experimental results show that the proposed framework achieves state-of-the-art performance on the 2015 MICCAI GlaS Challenge dataset and on a second independent colorectal adenocarcinoma dataset.

## 2 Methods

### 2.1 Minimal Information Loss Dilated Network

Gland instance segmentation is a complex task that requires a significantly deep network for meaningful feature extraction. Therefore, we use residual units to allow efficient gradient propagation through our deep network architecture. Traditional convolutional neural networks use a combination of max-pooling and convolution in a hierarchical fashion to increase the size of the receptive field [13]. The inclusion of max-pooling results in the loss of information with low activations, that may be very important for precise segmentation. To counter this loss of information, in addition to using traditional residual units, we include two additional types of residual unit during feature extraction:

minimal information loss (MIL) units and dilated residual units. The MIL unit incorporates the original image into each residual unit directly after the max-pooling layer. First, the original image is downsampled to the same size as the output of the pooling operation and then a 3×3 convolution is applied before concatenating to the output of the pooling layer. Next, a 3×3 convolution is applied to the concatenated block and this output is subsequently used in the residual summation operation as opposed to the input tensor in traditional methods. Three MIL units are added during feature extraction immediately after max-pooling. These MIL units can be seen in more detail within part (a) of Figure 2. A traditional residual unit can be defined as:

$$\mathbf{y} = \mathcal{F}(\mathbf{x}, \mathbf{W}_i) + \mathbf{x} \tag{1}$$

where $\mathbf{x}$ and $\mathbf{y}$ denote the input and output vectors respectively and $\mathbf{W}_i$ denotes the weights. Specifically $\mathcal{F}$ represents the function $\mathbf{W}_2(\sigma(\mathbf{W}_1\mathbf{x}))$, where $\sigma$ denotes ReLU. The addition of the the input vector $\mathbf{x}$ to $\mathcal{F}$ is shown by the summation operator $\oplus$ in the residual unit of part (c) in Figure 2. Equation (1) is modified to generate the MIL unit. The MIL unit can be defined as:

$$\mathbf{y} = \mathcal{F}(\mathbf{x}, \mathbf{W}_i) + \mathcal{G}(\mathbf{x}, \mathbf{v}, \mathbf{W}_j) \tag{2}$$

where $\mathcal{F}$ is defined in the same way as equation (1). The vector $\mathbf{v}$ denotes the original downsampled image and is incorporated into the function $\mathcal{G}$ to minimise the loss of information. $\mathcal{G}$ represents the function $\mathbf{W}_4(\sigma(\mathbf{W}_3\mathbf{v})\|\mathbf{x})$, where $\|$ denotes the concatenation operation. The summation of $\mathcal{F}$ and $\mathcal{G}$ is shown by the $\oplus$ symbol in the MIL unit within Figure 2.

Instead of downsampling the size of the input to increase the size of the receptive field, an alternate solution is to increase the size of the kernel during convolution. However, doing so is not feasible due to the huge amount of parameters required. Instead, dilated convolution uses sparse kernels [21], such that the resolution of the original image is preserved, without significantly increasing the number of parameters. We incorporate dilated convolution into residual units simply by replacing each 3×3 convolution with a 3×3 dilated convolution. We choose to initially downsample using max-pooling and MIL units because otherwise, convolving over the size of the original image for a sufficiently deep network, will lead to a blow up in the number of parameters. Minimising the loss of information allows us to perform a successful gland instance segmentation, without the need to incorporate additional information that is used in other methods [4]. It must be noted that we output the contours for uncertainty map refinement; not for separating gland instances. This is explained further in section 2.2. Dilated residual units can be seen in part (b) of Figure 2.

In addition, for effective multi-level aggregation, we apply atrous spatial pyramid pooling (ASPP) [5] to the output of the deep network with varying rates of dilation. In particular, within our framework the goal of ASPP is to combat the challenge of detecting glands of different cancer grades that show high variability in their size. When the dilation rate is too high, the dilated convolution operation reduces to a 1×1 convolution. This is because the dilated kernel becomes larger than the input feature map. Instead, to incorporate global level context, we also use global average pooling. All operations are followed by an initial 1×1 convolution, a dropout layer and then a second 1×1 convolution for reducing the depth of the output. The concatenation of these feature maps gives a powerful representation of the features extracted from the minimal information loss dilated network.

Although high-level contextual information can be generated within the deep neural network, it is crucial to incorporate low-level information for precisely delineating the glandular boundaries. Directly upsampling by a factor of 8 to produce the output does not consider low-level information. Instead, similar to U-Net [16], we choose to upsample by a factor of 2 each time and concatenate low-level features to the start of each upsampling block. This concatenation is shown by the dotted lines within Figure 2. Before the concatenation, we apply a 1×1 convolution to increase the depth of lower levels; ensuring that we have an equal contribution of both components during the concatenation. We find that this method of upsampling is especially important for precisely locating the boundaries where low-level features are particularly important. The overall flow of the feature extraction component of the network can be seen in Figure 2. We add deep supervision to our network by calculating the auxiliary loss at two points during feature extraction. This helps the network to learn more discriminative features and encourages a faster convergence.

During training, our overall loss function to be minimised is defined as:

$$\mathcal{L}_{total} = \sum_{a=1}^{2} w_a \mathcal{L}_a + \mathcal{L}_o + \mathcal{L}_c + ||\mathbf{W}||_2^2 \tag{3}$$

where $\mathcal{L}_a$ represents the auxiliary loss with corresponding discount weights $w_a$ that decay the contribution of the auxiliary loss during training. Auxiliary loss $\mathcal{L}_1$ defines the loss with respect to the gland object, whilst auxiliary loss $\mathcal{L}_2$ defines the loss with respect to the gland contour. We initially set $w_a$ as 1, and divide the value by 10 after every 8th training epoch. $\mathcal{L}_o$ and $\mathcal{L}_c$ represent the loss with respect to the gland object and gland contour at the output of the proposed network. $||\mathbf{W}||_2^2$ denotes the regularisation term on weights $\mathbf{W} = \{\mathbf{W}_a, \mathbf{W}_o, \mathbf{W}_c\}$. We define the cross-entropy loss $\mathcal{L}_a$, $\mathcal{L}_o$ and $\mathcal{L}_c$ as:

$$\mathcal{L}_a = \sum_{\mathbf{x} \in \chi} \log p_a(\mathbf{x}; \mathbf{W}_a); \quad \mathcal{L}_o = \sum_{\mathbf{x} \in \chi} \log p_o(\mathbf{x}; \mathbf{W}_o); \quad \mathcal{L}_c = \sum_{\mathbf{x} \in \chi} \log p_c(\mathbf{x}; \mathbf{W}_c) \tag{4}$$

where $p_a(\mathbf{x}; \mathbf{W}_a)$, $p_o(\mathbf{x}; \mathbf{W}_o)$ and $p_c(\mathbf{x}; \mathbf{W}_c)$ is the softmax classification at the auxiliary, object and contour output on input $\mathbf{x}$ in image space $\chi$, respectively.

## 2.2 Random Transformation Sampling for Uncertainty Quantification

Current deep learning models have an ability to learn powerful feature representations and are capable of successfully mapping high dimensional input data to an output. However, this mapping is assumed to be accurate in such models and there is no quantification of how certain the model is of the prediction. Within machine learning, a bayesian approach is often preferred, but traditional deep learning methods fail to successfully represent the uncertainty of a prediction. Recent work has aimed to quantify model uncertainty by finding the posterior distribution over the weights $P(\mathbf{W}|\mathbf{x}, \mathbf{y})$, where $\mathbf{x}$ is our observed input data and $\mathbf{y}$ is our set of labels. To estimate this posterior distribution, dropout variational inference [8, 11] can be used. However, this method uses dropout at multiple layers and this additional regularisation may have an adverse effect on the overall performance. Furthermore, model uncertainty can be reduced given enough data and therefore does not account for difficult cases irrespective of the amount of data that we have. Instead, we capture uncertainty by performing random transformations to the input images during test time. This allows us to capture the noise inherent in the observations [12] and allows us to visualise areas that are sensitive to small perturbations in the input space. To obtain the predictive distribution, we apply a random transformation $\Phi(\mathbf{x})$ on a sample of $n$ images, where $\Phi$ performs a flip, rotation, Gaussian blur, median blur or adds Gaussian noise on input image $\mathbf{x}$ to obtain $\{\Phi_1, \Phi_2, ..., \Phi_n\}$. Each image within the sample is then processed, where the mean of this processed sample gives the refined prediction and the variance gives the uncertainty. Concretely, we can define the prediction and uncertainty as:

$$\mu = \frac{1}{n} \sum_{i=1}^{n} f(\Phi_i(\mathbf{x}); \mathbf{W}); \quad \sigma = \frac{1}{n} \sum_{i=1}^{n} (f(\Phi_i(\mathbf{x}); \mathbf{W}) - \mu)^2 \tag{5}$$

where $\mu$ defines the segmentation prediction, $\sigma$ defines the uncertainty and $n$ defines the number of transformations. The function $f$ denotes the deep neural network with input $\mathbf{x}$ and output taken after the softmax layer. $\mathbf{W}$ denotes the weights and $\Phi_i$ defines a random transformation $i$ to input image $\mathbf{x}$

We propose a metric to give individual glands a score of uncertainty, based on the uncertainty map generated via random transformation sampling. This measure highlights glands that are generally hard to classify, irrespective of the number of examples. We suggest that it is reasonable to disregard segmented glands that have an uncertainty score above a given threshold, because in practice features would not be extracted from areas of general ambiguity. We first remove the boundaries by subtracting the predicted contours that have been output by the network and then calculate the object-level uncertainty score for each predicted instance $k$ as: $\tau_k = \frac{1}{n} \sum_{i=1}^{n} \hat{\sigma} \rho_{k,i}$, where $\hat{\sigma}$ is the boundary removed uncertainty map and $\rho_{k,i}$ is the predicted binary output of pixel $i$ within instance $k$. We define $n$ as the number of pixels within predicted instance $k$. We remove the boundaries because these areas show the transition between the two classes and therefore the uncertainty here can't be avoided. Given a selected global threshold for our uncertainty score $\tau$, we may only consider segmented glands with a score above this threshold.

# 3 Experiments and Results

## 3.1 Dataset and Pre-processing

For our experiments, we used two datasets: (i) the Gland Segmentation (GlaS) challenge dataset [17], used as part of MICCAI 2015, and (ii) a second independent colon adenocarcinoma dataset, which for simplicity we refer to as the colorectal adenocarcinoma gland (CRAG) dataset, that was originally used in [2]. Both datasets were obtained from the University Hospitals Coventry and Warwickshire (UHCW) NHS Trust in Coventry, United Kingdom. Within (i), there is a total of 165 image tiles taken from 16 H&E stained histological sections at $20\times$ magnification. The dataset consists of 85 training (37 benign and 48 malignant) and 80 test images (37 benign and 43 malignant). Furthermore, the test images are split into an off-site set A and an on-site set B. Images are mostly of size $775\times522$ pixels and all training images have associated instance-level segmentation ground truth that precisely highlight the gland boundaries. Within (ii), we have a total of 213 H&E CRA images taken from 38 WSIs, all of which are from different patients. Images are at $20\times$ magnification and are mostly of size $1512\times1516$ pixels, with corresponding instance-level ground truth. The CRAG dataset is split into 173 training images and 40 test images with different cancer grades. Examples of images from each of the two datasets can be seen in Figure 1.

We extracted patches of size $500\times500$ and augmented patches with elastic distortion, random flip, random rotation, Gaussian blur, median blur and colour distortion. Finally, we randomly cropped a patch of size $464\times464$, before input into the proposed network.

## 3.2 Implementation Details

We implemented our framework with the open-source software library TensorFlow version 1.3.0 [1]. The model was initialised with Gaussian distribution. We trained our model on a workstation equipped with one NVIDIA GEFORCE Titan X GPU for 30 epochs on the GlaS dataset and 75 epochs on the CRAG dataset. We used Adam optimisation with an initial learning rate of $10^{-4}$ and a batch size of 2.

## 3.3 Evaluation and Comparison

Table 1: Performance on the GlaS challenge dataset

| | $F_1$ Score | | | | Object Dice | | | | Object Hausdorff | | | | Rank |
|---|---|---|---|---|---|---|---|---|---|---|---|---|---|
| | A | Rank | B | Rank | A | Rank | B | Rank | A | Rank | B | Rank | Sum |
| **Proposed** | 0.914 | 1 | 0.844 | 1 | 0.913 | 1 | 0.836 | 1 | 41.54 | 1 | 105.89 | 1 | 6 |
| Xu et al. (b) [20] | 0.893 | 5 | 0.843 | 2 | 0.908 | 2 | 0.833 | 2 | 44.13 | 2 | 116.82 | 2 | 15 |
| MIMO-Net [15] | 0.913 | 2 | 0.724 | 7 | 0.906 | 3 | 0.785 | 9 | 49.15 | 4 | 133.98 | 6 | 31 |
| Xu et al. (a) [19] | 0.858 | 11 | 0.771 | 3 | 0.888 | 5 | 0.815 | 3 | 54.20 | 5 | 129.93 | 5 | 33 |
| DeepLab-v3 [5] | 0.862 | 10 | 0.764 | 5 | 0.859 | 13 | 0.804 | 5 | 65.72 | 9 | 124.97 | 4 | 46 |
| SegNet [3] | 0.858 | 11 | 0.753 | 6 | 0.864 | 12 | 0.807 | 4 | 62.62 | 10 | 118.51 | 3 | 46 |
| FCN-8 [14] | 0.783 | 14 | 0.692 | 12 | 0.795 | 14 | 0.767 | 10 | 105.04 | 12 | 147.28 | 9 | 71 |
| CUMedVision2 [4] | 0.912 | 3 | 0.716 | 9 | 0.897 | 4 | 0.781 | 11 | 45.42 | 3 | 160.35 | 13 | 43 |
| ExB1 | 0.891 | 7 | 0.703 | 10 | 0.882 | 8 | 0.786 | 7 | 57.41 | 10 | 145.58 | 7 | 49 |
| ExB3 | 0.896 | 4 | 0.719 | 8 | 0.886 | 6 | 0.765 | 13 | 57.36 | 9 | 159.87 | 12 | 52 |
| Freidburg2 [16] | 0.87 | 8 | 0.695 | 11 | 0.876 | 9 | 0.786 | 7 | 57.09 | 7 | 148.47 | 10 | 52 |
| CUMedVision1 [4] | 0.868 | 9 | 0.769 | 4 | 0.867 | 11 | 0.8 | 6 | 74.6 | 13 | 153.65 | 11 | 54 |
| ExB2 | 0.892 | 6 | 0.686 | 13 | 0.884 | 7 | 0.754 | 14 | 54.79 | 6 | 187.44 | 14 | 61 |
| Freidburg1 [16] | 0.834 | 13 | 0.605 | 14 | 0.875 | 10 | 0.783 | 10 | 57.19 | 8 | 146.61 | 8 | 63 |
| CVML | 0.652 | 16 | 0.541 | 15 | 0.644 | 17 | 0.654 | 15 | 155.43 | 17 | 176.24 | 14 | 94 |
| LIB | 0.777 | 15 | 0.306 | 17 | 0.781 | 15 | 0.617 | 16 | 112.71 | 16 | 190.45 | 16 | 95 |
| vision4GlaS | 0.635 | 17 | 0.527 | 16 | 0.737 | 16 | 0.61 | 17 | 107.49 | 15 | 210.1 | 17 | 98 |

Table 2: Performance on the CRAG dataset

| | $F_1$ Score | | Object Dice | | Object Hausdorff | | Rank |
|---|---|---|---|---|---|---|---|
| | Score | Rank | Score | Rank | Score | Rank | Sum |
| **Proposed** | 0.825 | 1 | 0.875 | 1 | 160.14 | 1 | 3 |
| DCAN [4] | 0.736 | 2 | 0.794 | 2 | 218.76 | 2 | 6 |
| DeepLab-v3 [5] | 0.648 | 3 | 0.745 | 3 | 281.45 | 4 | 10 |
| SegNet [3] | 0.622 | 4 | 0.739 | 4 | 247.84 | 3 | 11 |
| U-Net [16] | 0.600 | 5 | 0.654 | 5 | 354.09 | 5 | 15 |
| FCN-8 [14] | 0.558 | 6 | 0.640 | 6 | 436.43 | 6 | 18 |

Table 3: Performance with random transformation sampling (RTS) on both the GlaS and CRAG dataset.

| | $F_1$ Score | | | Object Dice | | | Object Hausdorff | | |
|---|---|---|---|---|---|---|---|---|---|
| | GlaS A | GlaS B | CRAG | GlaS A | GlaS B | CRAG | GlaS A | GlaS B | CRAG |
| Proposed | 0.914 | 0.809 | 0.806 | 0.908 | 0.822 | 0.867 | 42.32 | 117.91 | 162.35 |
| Proposed-RTS | 0.914 | **0.844** | **0.825** | **0.913** | **0.836** | **0.875** | **41.54** | **105.89** | **160.14** |

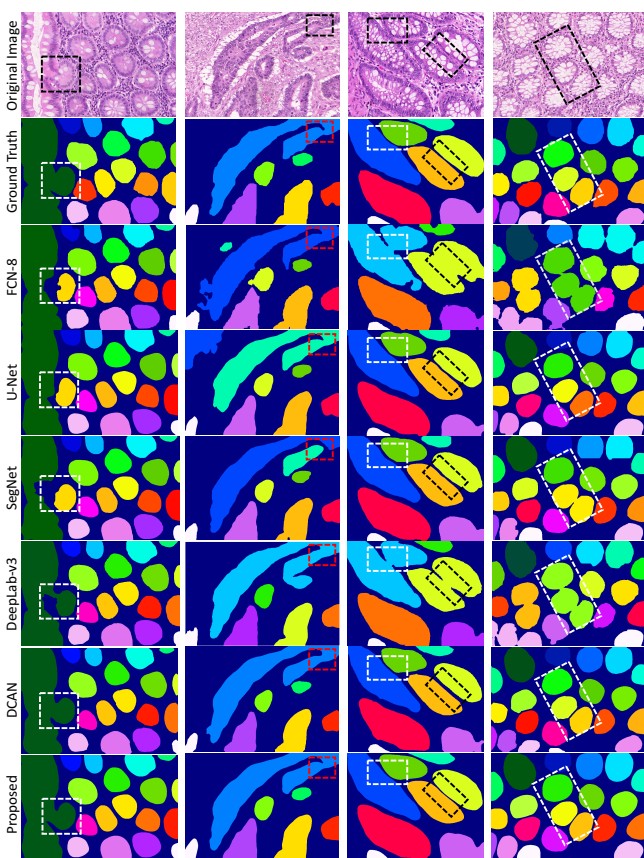

Figure 3: Visual gland segmentation results on the GlaS dataset. We compare our method to state-of-the-art methods including FCN-8, U-Net, SegNet, DCAN and DeepLab-v3. Note, visual results for U-Net and DCAN are the results as submitted to the GlaS challenge.

We assessed the performance of our algorithm by using the same evaluation criteria used in the MICCAI GlaS challenge, consisting of $F_1$ score, object-level dice and object-level Hausdorff distance [17]. Furthermore, we implemented several state-of-the-art segmentation methods including SegNet [3], FCN-8 [14] and a DeepLab-v3 [5] model for extensive comparative analysis. We also report the results obtained by two recent methods including MIMO-Net [15], that uses a multi-input-multi-output convolutional neural network and two methods that utilise deep multichannel side supervision [19, 20]. We can see that our proposed network achieves state-of-the-art performance compared to all methods on the 2015 MICCAI GlaS Challenge dataset within Table 1. We also validated the efficacy of our method on the CRAG dataset, demonstrating overall better performance in comparison with other methods and highlighting the good generalisation capability of our method on different datasets. Results on the CRAG dataset can be seen in Table 2. It is interesting to see that within the dashed boxes in the last column of Figure 4, our proposed algorithm was able to detect tumorous areas that were not picked up by the pathologist. We can see from Table 3 that utilising test time random transformations leads to an improved performance, due to a refined prediction within areas of high uncertainty. It must be noted that it is significantly more difficult to segment glands within the CRAG dataset than when using the GlaS dataset. This is because there are many malig-

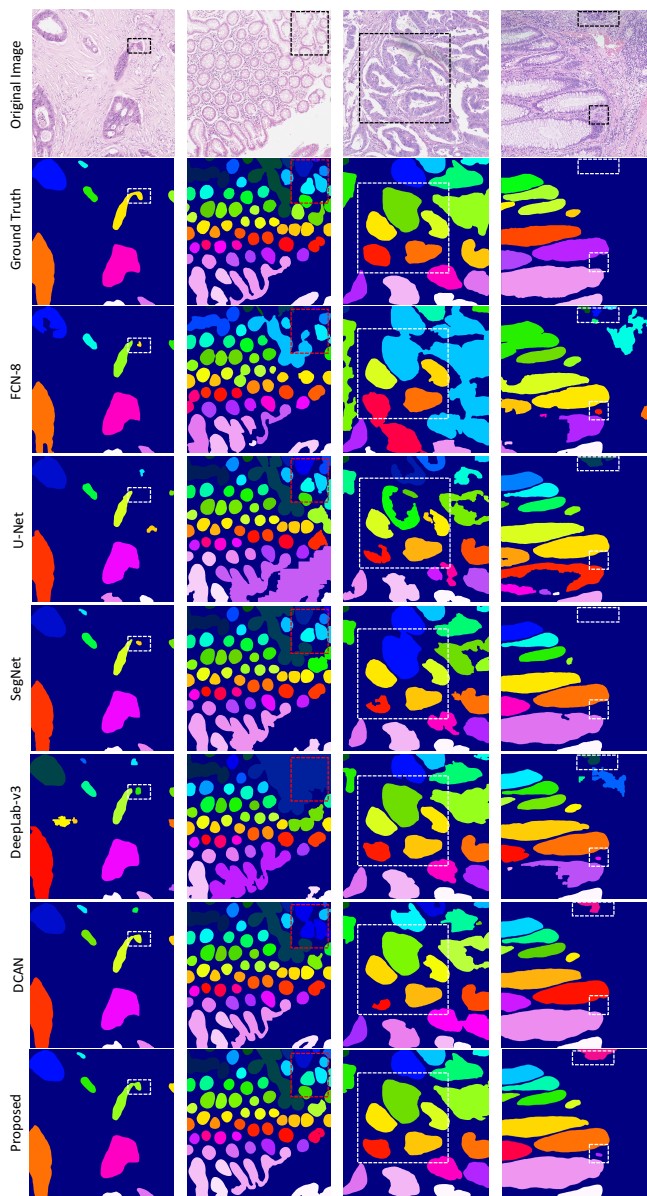

Figure 4: Visual gland segmentation results on the CRAG dataset. We compare our method to state-of-the-art methods including FCN-8, U-Net, SegNet, DCAN and DeepLab-v3.

nant cases where the glandular boundaries are very ambiguous. Examples of results from different methods are shown in Figure 3 and 4. We can see that our method can generate more accurate gland instance segmentation with precisely delineated boundaries and well segmented instances. In Figure 5, we show the relationship between the performance and the uncertainty score $\tau$. This score is used as a threshold, where we only consider predictions $k$ with an uncertainty score $\tau_k$ lower than $\tau$. We observe from Figure 5 that it seems sensible to only consider segmented predictions with an uncertainty score $\tau_k$ below 1. This preserves a large proportion of the dataset, whilst significantly increasing the performance. It is interesting to note that we are still able to preserve around 75% of instances by selecting predictions with $\tau_k$ below 0.25. As a result, $F_1$ score, object dice and object Hausdorff can be increased to 0.930, 0.9359 and 28.658 for test set A and increased to 0.913, 0.9567 and 22.70 for test set B.

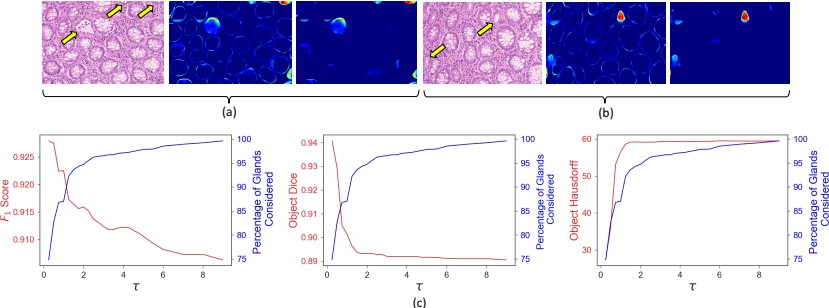

Figure 5: Object-level uncertainty quantification. (a,b) from left to right: original image; uncertainty map $\sigma$; boundary removed uncertainty map $\hat{\sigma}$. For each instance $k$ within $\hat{\sigma}$, an object-level uncertainty score $\tau$ is calculated. (c) shows the $F_1$ score, object dice and object Hausdorff as we disregard predictions with an uncertainty score $\tau_k$ greater than a given threshold $\tau$. For each graph, we show the percentage of total instances considered, given a threshold $\tau$. Graphs within (c) relate to results on the combined set of test A and test B. Yellow arrows highlight areas with a high uncertainty.

## 4   Conclusion

In this paper, we presented a minimal information loss dilated network for gland instance segmentation in colon histology images. The proposed network retains maximal information during feature extraction that is very important for successful gland instance segmentation. Furthermore, in order to segment glands of various sizes, we use atrous spatial pyramid pooling for effective multi-scale aggregation. To incorporate uncertainty within our framework, we apply random transformations to images during test time. Taking the average of this sample leads to a superior segmentation, whilst simultaneously allowing us to visualise areas of ambiguity. Furthermore, we propose an object-level uncertainty score that can be used for assessing whether to discard predictions with high uncertainty. We observe that our method obtains state-of-the-art performance in the MICCAI 2015 gland segmentation challenge and on a second independent colorectal adenocarcinoma dataset.

## 5   Acknowledgements

The authors are grateful to the Warwick Global Partnership Fund (GPF) 2017/18 for funding this collaboration between Warwick and CUHK. H.C, Q.D and P.-A.H are supported by the Hong Kong Innovation and Technology Commision, under ITSP Tier 3 (project number: ITS/041/16). We thank Jevgenij Gamper for fruitful discussions relating to uncertainty quantification within medical images.

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
