# OpenReview forum: "MILD-Net: Minimal Information Loss Dilated Network for Gland Instance Segmentation in Colon Histology Images"
_MIDL.amsterdam/2018/Conference — MIDL 2018 Oral_

### Review · AnonReviewer3 · 2018-05-06
**This work could be a reasonable way to make the performance of segmentation task better.**

**Rating:** 5
**Confidence:** 3

**Review:**

This work represented a novel method to make the performance of segmentation task better by using minimal information loss dilated network and Monte Carlo dropout sampling.

1. Can you describe existing methods during testing to compare with Monte Carlo dropout sampling?
2. Haven't you every considered any other methods for color normalization?
3. Can you test other dataset from other hospital for extra-validation?
4. Please identify SD of accuracies in Table 2 and 3?
5. Please describe implementation details more to get the best deep learning model in training.
6. Please correct "Minimial" to "minimal" in the title.

**Special Issue:**

Definitely

---

### Review · AnonReviewer1 · 2018-05-09
**A novel and interesting paper**

**Rating:** 4
**Confidence:** 3

**Review:**

Overall:
The paper proposes a new architecture for gland segmentation. I find the proposed approach very interesting and novel. The obtained results confirm the superiority of the proposed approach over other methods.

Strengths:
+ The paper is very well written.
+ The experiments are very well carried out.
+ The application of dilated convolutions and atrous spatial pyramid pooling is well motivated.
+ The application of the MC dropout is not new but it makes a lot of sense in the considered problem.
+ The results presented in the paper are very convincing.
+ The discussion of related work is very thorough and up-to-date.

Remarks:
* Minor
- The authors claim that the application of the residual units alongside with dilated convolutions help to retain maximal information. I understand the motivation for that, however, this is pretty misleading because the authors do not show any properties of the proposed minimal information loss units in terms of the mutual information or any other information-theoretic measure.
- The proposed architecture is extremely complicated. I am fully aware of resource limitations in academia, however, did the authors check less complicated architectures by removing some layers? If so, how did it influence the final performance?

**Special Issue:**

Definitely

---

### Review · AnonReviewer2 · 2018-05-10
**Well designed study with a novel architecture**

**Rating:** 4
**Confidence:** 2

**Review:**

This paper proposes a novel deep learning architecture and applies that method to the task of colon gland segmentation. The presented method is compared to numerous state of the art approaches and is shown to outperform them. The paper is well written with descriptive and clear figures. I recommend acceptance of this study.

Strengths:
1. This study uses appropriate statistical analysis techniques to evaluate their method's performance.
2. This study uses two datasets with differing pathological content.
3. The use of a challenge dataset allows for logical and clear comparison to existing methods.
4. The demonstrated method outperforms existing methods and the authors clearly explain where this performance gain was achieved (glands in close proximity).

Suggestions:
1. The Monte Carlo Dropout sampling improved performance by some measures in some datasets but not in others. The discussion of why this was the case with Hausdorff distance was interesting and an expanded discussion on this topic could strengthen the paper.
2. The benefits of the Monte Carlo Dropout method appear to be small (on the order of a 1% improvement). Are these performance gains significant?
3. The model was trained for a long time (70 epochs, 75 hours). Was the long training needed to achieve this performance or could an earlier epoch have been used?

**Special Issue:**

Definitely

---

### Comment · ~Bram_van_Ginneken1 · 2018-05-18
**Selection for longlist for special issue Medical Image Analysis**

Dear authors,

Congratulations on your acceptance to MIDL! We have selected your paper on the longlist for the Medical Image Analysis Special Issue. Please read this page:
https://midl.amsterdam/special-issue-in-medical-image-analysis/
Please answer the three questions that are listed on that page about your interest in submitting to the special issue, potential overlap with other publications, and related publications.

You can post your answer here directly below on openreview.net, or mail me directly at bram.vanginneken@radboudumc.nl.

Best regards, Bram

---

### Decision · Program_Chairs · 2018-05-15
**Paper38 Acceptance Decision**

Oral